# Sex Maintenance in Mammals

**DOI:** 10.3390/genes12070999

**Published:** 2021-06-29

**Authors:** Rafael Jiménez, Miguel Burgos, Francisco J. Barrionuevo

**Affiliations:** Centro de Investigación Biomédica, Lab. 127, Departamento de Genética, Instituto de Biotecnología, Universidad de Granada, 18016 Granada, Spain; rjimenez@ugr.es (R.J.); mburgos@go.ugr.es (M.B.)

**Keywords:** mammalian sex maintenance, sex determination, testis differentiation, ovary differentiation, gonadal cells transdifferentiation, gonadal genetic reprograming

## Abstract

The crucial event in mammalian sexual differentiation occurs at the embryonic stage of sex determination, when the bipotential gonads differentiate as either testes or ovaries, according to the sex chromosome constitution of the embryo, XY or XX, respectively. Once differentiated, testes produce sexual hormones that induce the subsequent differentiation of the male reproductive tract. On the other hand, the lack of masculinizing hormones in XX embryos permits the formation of the female reproductive tract. It was long assumed that once the gonad is differentiated, this developmental decision is irreversible. However, several findings in the last decade have shown that this is not the case and that a continuous sex maintenance is needed. Deletion of *Foxl2* in the adult ovary lead to ovary-to-testis transdifferentiation and deletion of either *Dmrt1* or *Sox9/Sox8* in the adult testis induces the opposite process. In both cases, mutant gonads were genetically reprogrammed, showing that both the male program in ovaries and the female program in testes must be actively repressed throughout the individual’s life. In addition to these transcription factors, other genes and molecular pathways have also been shown to be involved in this antagonism. The aim of this review is to provide an overview of the genetic basis of sex maintenance once the gonad is already differentiated.

## 1. Testis and Ovary Cell Lineages Originate from the Same Precursors

Although adult testes and ovaries have equivalent functions, that is, the production of both gametes and sex hormones, their anatomy and genetic programs are very different. Nevertheless, both organs share the same ontogenetic origin. In the testis, gamete production takes place in the seminiferous tubules, which are formed by the so called “supporting cells”, the Sertoli cells, whose nuclei occupy basal positions in the germinative epithelium of these tubules, and the germ cells, which differentiate from spermatogonial stem cells into spermatozoa through spermatogenesis [1]. Sertoli cells nurse and regulate germ cell function and provide them with the required nutrients and structural support for male gamete generation [2,3]. Steroidogenic Leydig cells, whose main function is the production of androgens, are located in the interstitial space among seminiferous tubules [4]. The adult ovary displays a completely different tissue organization to fulfill similar functions. Germ cells (oocytes arrested at the first meiotic prophase) are surrounded by the supporting cell line, the granulosa cells, forming the primordial follicles that grow to release mature oocytes [5]. Associated with growing follicles are also the endocrine theca cells [6]. All these cell types differentiate from the bipotential gonad, an embryonic structure with the capability to differentiate into either a testis or an ovary. Classical studies using XX-XY chimeric mouse testes showed that Sertoli cells were predominantly XY, whereas XX cells contributed mainly to other cell types including Leydig cells, peritubular myoid cells, and vascularized connective tissue, indicating (1) that the supporting cell lineage of the testis, the Sertoli cells, is the first cell type in the gonad to follow a sex-specific fate, and (2) that subsequent steps in testis differentiation is a consequence of Sertoli cell activity [7]. Further experiments using transgenic mice expressing the testis determining gene, *Sry*, to trace the fate of supporting cells showed that XX and XY supporting cell lines have a common precursor [8,9]. This “common progenitor identity” hypothesis was later confirmed in studies using transcriptomic profiling [10,11,12,13]. Using lineage-specific transgenic mice, Jameson et al. (2012) showed that the supporting cells appeared to acquire their sex-specific fates by embryonic day 11.5 (E11.5), whereas the sex-specific differentiation of both interstitial/stromal cells and germ cells began at E12.5, when the supporting cells had almost completed their process of differentiation. This provides evidence at the transcriptome level that the supporting cells of both sexes are the first cell types in the gonad to follow a sex-specific fate. More recently, single cell RNA sequencing (scRNA-seq), which allows for the recapitulation of the differentiation process of each cell type in a heterogeneous population, has evidenced the absence of a cell lineage-specific fate prior to sex determination. It also revealed that the first event of gonad differentiation is the adoption of a supporting-cell fate by multipotent progenitors with similar transcriptomic profiles in both XX and XY gonads, followed by the sex-specific differentiation into either pre-Sertoli or pre-granulosa cells. The rest of the progenitor cells that give rise to different cell populations including the steroidogenic Leydig cells in the testis and theca cells in the ovary experience transcriptomic changes with a patent sexual dimorphism at later stages [14,15].

The molecular mechanisms underlying sex determination and early gonad differentiation have been extensively reviewed and updated in recent years [16,17,18,19,20,21,22,23]. Here, we will highlight only the most relevant events, focusing on genes whose function in the adult gonads will be discussed hereafter. As above-mentioned, sex differentiation starts with the specification of the supporting cell progenitors that differentiate as either Sertoli cells in the testis or granulosa cells in the ovary. In the XY gonad, the expression of the Y-linked, testis determining gene, *SRY*, in pre-Sertoli cells leads to the upregulation of *SOX9*, which in turn activates a cascade of male promoting genes including *DMRT1*, *AMH*, and *SOX8* that orchestrate Sertoli cell differentiation. Subsequently, Sertoli cells undergo a mesenchymal to epithelial transition and form the testis cords that enclose germ cells. Sertoli cells also promote the differentiation of the steroidogenic Leydig cells and prevent germ cells from meiosis entry. In mice, all these events are almost completed within 24–48 h after sex determination. In contrast, cell lineage differentiation is slower in the ovary. The undifferentiated XX supporting precursors lack *SRY* and the WNT signaling genes *WNT4* and *RSPO1* are upregulated. Once committed to the granulosa cell fate, other genes and pathways such as *FOXL2*, *TGFβ*, and *FST* are also upregulated. In mice, the process of granulosa cell differentiation occurs over several days, and is completed after birth, when folliculogenesis starts [21]. In contrast to the testis, ovarian germ cells enter meiosis during embryonic development, and primary follicles, in which germ cells are surrounded by a monolayer of granulosa cells, are observed after birth. In the adult testis, solid cords have been substituted by seminiferous tubules with a lumen and a thick germinative epithelium in which a variety of mitotic, meiotic, and post-meiotic germ cells are present in different stages of maturation including spermatogonia, spermatocytes, spermatids, and sperm. Leydig cells maintain a high level of steroidogenic activity. In the adult ovary, follicles are seen in different sizes according to their degree of maturation. In growing follicles, steroidogenic theca cells surround the granulosa cells which, in turn, surround the oocyte (Figure 1).

## 2. Plasticity of the Gonadal Cell Fates after Sex Determination

The assumption that gonadal differentiation is irreversible was motivated by the fact that most cases of XX or XY sex reversal could be explained by functional failure of sex-specific factors at the sex determination stage, suggesting that alterations were never produced at later stages. However, cases of transdifferentiation between gonadal sex-specific cell lines, mainly the supporting cell line, after sex determination have long been described. For instance, the ovary of old rats contained structures resembling testis cords that were not present in young rats [24]. Additionally, several cases were reported in which testis-like structures developed in the ovary after the loss of the meiotic germ cells. E12.0 mouse ovaries transplanted beneath the kidney capsules of adult male mice initially developed as ovaries, but seminiferous cords with Sertoli-like cells and testosterone-producing Leydig cells started to develop from the twelfth day after transplantation in regions depleted of oocytes [25]. Similarly, E14.5 rat ovaries cultured in a medium conditioned by either fetal or young testes, lost germ cells, and developed testis-like cords after 12 days of culture [26], and undifferentiated tammar wallaby ovaries transplanted under the skin of young male pouch also became depleted of germ cells and contained seminiferous-like cords 25 days after transplantation [27]. In addition, granulosa cells survive, proliferate, and subsequently acquire morphological characteristics of Sertoli cells in rat ovarian follicles depleted of oocytes by irradiation [28]. We also showed that after depletion of germ cells in the medullary region of the developing XX gonad of the Iberian mole, *Talpa occidentalis*, testis cord-like structures are formed and Leydig cells subsequently appear in the interstitial spaces [29].

Another case of granulosa-to-Sertoli cell transdifferentiation was observed in freemartinism, a syndrome in which XX female cattle fetuses with male twins exhibit female-to-male sex reversal with varying degrees of female reproductive tract misdevelopment [30]. In about 50% of the cases, the XX gonads present seminiferous-like tubules [30], and in the most severe cases, Leydig-like cells were also described [31]. Freemartinism occurs when chorionic vascular anastomosis allows the transfer of male gonad derived hormones to the female twin embryo, affecting its sexual development. Because the male reproductive ducts (Müllerian ducts) regress at the same time in the freemartin female as in her male twin, it was proposed that the Anti-Müllerian hormone (AMH, also known as Müllerian inhibiting substance, MIS), a member of the transforming growth factor β family produced by Sertoli cells that induces the regression of Müllerian ducts in male fetuses [32], could be the factor responsible for the syndrome. Indeed, in vitro culture of E14.5 rat ovaries in the presence of purified bovine, AMH showed a reduction of the gonadal volume, oocyte depletion, and differentiation of Sertoli-like cells [33]. Likewise, transgenic mice chronically expressing *Amh* presented ovaries with few germ cells at birth that were lost afterward, coinciding with the formation of seminiferous-like tubules [32]. The molecular mechanism underlying this process remains unknown. It has been suggested that it is the loss of oocytes derived from the presence of AMH (this hormone is cytotoxic for oocytes), rather than the presence of AMH itself, which causes the transdifferentiation [34,35]. However, this is controversial as the effect of oocyte depletion on granulosa cell transdifferentiation is not well understood. It seems to depend on the stage of germ cell development, and does not occur if the germ cells are depleted before the pre-meiotic stages [36]. However, since postnatal oocyte depletion (like in ovaries exposed to AMH) leads to transdifferentiation in some cases [28], but not in others [37], it cannot be ruled out that AMH may have a direct role in the process of granulosa cell transdifferentiation.

Cell transdifferentiation has also been described in cases of human gonadal cancers. The Sertoli-Leydig cell tumor (SLCT) of the ovary is a rare type of tumor normally affecting middle-aged women, which is characterized by the presence of testicular structures including Sertoli-like cells and Leydig cells that produce androgens [38]. On the other hand, cases of granulosa-cell tumors have also been reported in which neoplastic proliferation of intratubular sex cord cells progresses to an invasive tumor, simultaneously experiencing granulosa cell differentiation and losing Sertoli cell features [39,40]. However, in both cases, supporting cell transdifferentiation may be just a secondary consequence of the dramatic alterations taking place in the genetic program of tumor cells.

Additional evidence of the plasticity of the gonadal cell fate came from transgenic mice. *Sry* ectopic expression in XX embryonic gonads using a heat-shock-inducible system revealed that *Sox9* expression was upregulated and maintained in pre-granulosa cells when *Sry* expression was induced during the E11.0–11.25 critical time window, but not afterward [41]. However, fine monitoring of granulosa cells in these mice revealed that the SRY-dependent *Sox9* inducibility was not as transient in a subpopulation of pre-granulosa cells near the mesonephric tissue, which maintained that capability throughout fetal and early postnatal stages. Furthermore, when E13.5 ovaries were grafted into adult male nude mice, the heat-shock *Sry* transgene was also able to induce *Sox9* expression in differentiated granulosa cells [42].

Finally, chromatin accessibility landscape analyses using purified Sertoli cells have shown that this cell type maintains open chromatin regions near female-promoting genes that are enriched in transcription factor binding motifs for male-promoting genes such as *Sox9* and *Dmrt1*, indicating that these genes are continuously acting as silencers by binding repressors of the granulosa cell fate [43,44]. ChIP-seq for H3K27me3 and H3K4me3 provided results consistent with this notion. H3K27me3 indicates promoter repression, whereas H3K4me3 evidences promoter activation. ChIP-seq experiments performed using purified adult Sertoli cells showed that male-promoting genes harbored H3K4me3 and were depleted in H3K27me3, whereas female-promoting genes were enriched for both marks, indicating that female-determining genes persist in a poised state even long after Sertoli cell differentiation [45]. These results indicate that cells of the supporting lineage require the opposite sex to be permanently repressed, and explain why the loss of such repressors can lead to transdifferentiation into the opposite cell lineage even long after the sex determination stage.

Cases of XX and XY sex reversal induced after sex determination have been associated with a number of sex-specific genes and pathways. We will review the most outstanding cases as follows.

## 3. Genes and Pathways Involved in the Maintenance of the Female Cell Fate

### 3.1. Inhibin/Follistatin/TGF-β

Activins and inhibins are dimeric proteins belonging to the TGF-β superfamily that stimulate and inhibit the pituitary FSH secretion, respectively [46]. They also regulate many other biological processes including germ cell development, follicle maturation, ovulation, and uterine receptivity [47]. Mice with a targeted deletion of *Inha*, the gene of Inhibin-α, which is the common monomeric subunit shared by Inhibin A and B proteins [46], were infertile and developed gonadal tumors in both males and females. In these mutant mice, initial gonadal development was normal but their ovaries presented nodules of seminiferous-like tubules with Sertoli-like cells at 6–8 weeks [48]. Follistatin (FST) is a glycoprotein that cooperates with inhibins in neutralizing activins and pituitary FSH repression [47,49,50]. Granulosa cell-specific deletion of *Fst* in mice also led to reduced fertility with adult ovaries showing some seminiferous-like testis cords [51]. In both *Inha* and *Fst* mutant mice, FSH levels were increased [48,51], indicating that high levels of this hormone could mediate the granulosa-to-Sertoli cell transdifferentiation displayed by both types of mutant ovaries. However, this does not seem to be the case, as the ovaries of transgenic mice overexpressing FSH showed no sign of transdifferentiation [52]. FST and inhibins regulate TGF-β/Activin signaling during foliculogenesis [49,50]. Interestingly, the activation of TGF-β receptor type-1 (TGFBR1) in Sertoli cells led to Inhibin A, FOXL2, and WNT signaling upregulation [53], thus exhibiting an evident granulosa-cell type gene expression pattern. Thus, proper regulation of TGF-β/Activin appears to be necessary for the maintenance of the female sexual fate, although the precise mechanisms governing these processes have yet to be determined.

### 3.2. Estrogen Signaling

Estrogen signaling regulates several physiological processes including normal cell growth, differentiation, and function of target tissues such as the reproductive tracts, the mammary gland, and central nervous and skeletal systems [54,55]. Estrogen action is mediated by binding of the hormone (E2 or 17β Estradiol) to one of its two nuclear receptors, the estrogen receptor α, ERα (gene name: *ESR1*) and the estrogen receptor β, ERβ (gene name: *ESR2*) [54,55]. Generation of null mutant mice for each receptor revealed that female *Esr1* mutants (*Esr1*KO) were sterile, whereas female mutants for *Esr2* (*Esr2*KO) presented variable degrees of subfertility [56,57,58]. The generation of double *Esr1* and *Esr2* null mutants (*Esr12*KO) revealed that folliculogenesis was impaired with antral follicles containing a very reduced number of granulosa cells. In addition, adult *Esr12*KO ovaries exhibited structures resembling testis tubules with Sertoli-like cells expressing *Amh* and *Sox9*. These structures were not visible in prepubertal double mutant mice, and the effect increased with age, indicating that a process of adult granulosa-to-Sertoli cell transdifferentiation had occurred [58,59].

A similar ovarian phenotype was described in mice with a null mutation in the aromatase gene (*Cyp19a1*; *ArKO*), an enzyme that catalyzes the last step of estrogen biosynthesis. Adult *Ar*KO ovaries presented increased expression of *Sox9* and developed seminiferous-like structures containing Sertoli-like cells and interstitial Leydig-like cells. Estrogen administration to mutant mice decreased the number of Sertoli and Leydig cells in the ovaries, confirming the essential role of estrogen signaling in the maintenance of the ovarian fate [60].

### 3.3. FOXL2

FOXL2 is a transcription factor belonging to the winged helix or Forkhead family whose mutation in humans leads to the autosomal dominant blepharophimosis/ptosis/epicanthus inversus syndrome (BPES), which affects the eyelids and the ovary [61]. The first evidence that this factor could play a role in mammalian sex determination came from the identification of a deletion of 11.7 kb in the sex-reversed polled goat, which included *FOXL2* [62]. A later study showed that *FOXL2* is a female sex-determining gene in goat [63]. In contrast, *FOXL2* is dispensable for mouse sex determination, although it has essential roles during ovarian development [16,20,22]. In 2009, Uhlenhaut and colleagues showed that conditional inactivation of *Foxl2* in the adult mouse ovary resulted in an immediate transdifferentiation of granulosa cells into Sertoli-like cells that formed testis-like seminiferous tubules expressing male promoting genes including *Sox9*. In addition, the steroidogenic theca cells were transformed into Leydig-like cells, which produced testosterone at levels similar to those of males. Interestingly, the authors showed that granulosa transdifferentiation occurred in the presence of oocytes and that ovaries depleted of germ cells do not undergo deregulation of the sex-specific factors including *Foxl2* and *Sox9*, proving that granulosa transdifferentiation is a direct consequence of the lack of *Foxl2* in granulosa cells and that it is independent of oocyte loss [37]. Several observations indicate that FOXL2 cooperates with the estrogen signaling pathway in the maintenance of the granulosa cell fate, (a) FOXL2 and ESR1 synergistically repress a cis-regulatory element required for testis-specific *Sox9* expression (TESCO) [37], (b) 8-week old ovaries of female mice heterozygous for both *Foxl2* and *Esr1* mutations contained Sertoli-like cells expressing SOX9, something that had not been observed in simple mutants [37], and (c) in mouse primary follicle cells, FOXL2 regulates *Esr2* expression and is required for normal gene regulation by steroid receptors [64]. RUNX1 is another factor that may cooperate with FOXL2 in preventing postnatal ovary masculinization as XX *Runx1/Foxl2* double mutant gonads showed partial masculinization at birth with structures similar to fetal testis cords that expressed *Dmrt1*, but not *Sox9*, an effect that was not observed in simple mutants. Furthermore, FOXL2 and RUNX1 exhibit overlaps in chromatin binding in fetal ovaries [65].

### 3.4. WNT Signaling

XX sex reversal in mice lacking *Wnt4* revealed an important role for *Wnt* signaling in ovary development [66]. Mutations of the *WNT* ligand gene, *Rspo1*, resulted in XX sex reversal [67], and testicular stabilization of β-catenin, the intracellular mediator of this pathway, lead to XY sex reversal [68], thus confirming the essential role for this genetic pathway in ovary differentiation. However, in both *Wnt4* and *Rspo1* mutant ovaries, granulosa cells initially differentiate and only shortly before birth, a partial female-to-male sex reversal occurs with the formation of some testis-like cords with Sertoli-like cells expressing *Sox9* [69,70]. As granulosa cells become proliferative prior to this cell transdifferentiation, Maatouk and colleagues hypothesized that transdifferentiation in the absence of *Wnt* signaling will only occur if granulosa cells are actively proliferating [70], a possibility that would also explain other cases of granulosa-to-Sertoli transformation either with or without oocyte loss [33,37,58,59,60,71]. In any case, it is clear that WNT signaling is involved in the maintenance of the granulosa-cell fate, as Sertoli cell-specific stabilization of β-catenin several days after the sex determination stage transforms Sertoli cells into granulosa-like cells. This process is mediated by *Foxl2*, whose expression is activated by the binding of β-catenin to the transcription factor *Tcf/Lef*-binding sites in the *Foxl2* promoter [72].

### 3.5. FOXO1 and FOXO3

The Forkhead box (FOX), FOXO1 and FOXO3, are transcription factors that regulate cellular differentiation, growth, survival, cell cycle, metabolism, and stress [73]. Murine granulosa cell-specific inactivation of both *Foxo1* and *Foxo3* impairs follicular development causing infertility. Ablation of the tumor suppressor gene *Pten* in the *Foxo1/3* mutant strain enhances the penetrance of this phenotype [74]. In double and triple mutants, SOX9 was detected in granulosa cells of follicles that had lost the oocyte as well as in tubular structures. However, expression profiling showed that other Sertoli-specific markers such as *Rhox8*, *Sox8*, and *Tsx* were not upregulated in triple *Foxo1/3/Pten* mutant granulosa cells, indicating that full granulosa-to-Sertoli cell transdifferentiation does not occur in the absence of the three factors.

### 3.6. Hippo Signaling

Hippo is a signaling pathway involved in cell fate determination, differentiation, proliferation, and apoptosis of many cell types during embryogenesis [75,76]. The kinases large tumor suppressors 1 and 2 (LATS1, LATS2) are two proteins involved in the cascade of phosphorylations that take place during hippo signal transduction [75,76]. Female mice mutants for *Lats1* were subfertile with ovaries containing a reduced number of antral follicles and no corpora lutea [77]. In these mice, no granulosa-to-Sertoli cell transdifferentiation was reported, although the ovarian parenchyma was transformed into bone tissue and seminiferous tubule-like structures in double *Lats1/2* mutant mice. In these mice, the nuclear regulator of hippo signaling Yes-associated protein (YAP) presented a markedly enhanced recruitment to the *Sox9* proximal promoter, particularly near the transcriptional start site, indicating that loss of *Lats1* and *2* increases YAP promoter binding activity, which leads to ectopic upregulation of genes that promote the male sexual fate [78].

## 4. Genes Involved in the Maintenance of the Male Cell Fate

### 4.1. DMRT1

The doublesex and mab-3 related transcription factor 1 (*DMRT1*) maps to a region in human chromosome 9p that, when deleted, causes defective testicular development and XY feminization [79]. Targeted deletion in mice showed that the gene is necessary for postnatal testis differentiation [80]. Mice with a Sertoli cell ablation of *Dmrt1* shortly after the sex determination stage presented a normal embryonic and postnatal testis development, but around postnatal day 14 (P14), the supporting cells started to express *Foxl2*, and by P28, most of the supporting cells expressed *Foxl2* but not *Sox9*. Transcriptomic analysis of P28 gonads revealed a sexual reprogramming of Sertoli cells into granulosa cells accompanied by theca cell formation, estrogen production, and feminization of germ cells. Furthermore, the inactivation of *Dmrt1* in adult testes revealed the presence of granulosa-like cells expressing FOXL2, confirming that *Dmrt1* is essential to maintain male cell identity in the mouse gonad throughout life [81]. Further studies revealed that DMRT1 antagonizes the feminizing action of retinoic acid (RA) signaling, a pathway that is necessary for male gametogenesis and for Sertoli cell function [82,83]. In Sertoli cells deficient for *Dmrt1*, RA activates female promoting genes including *Foxl2*, *Esr2*, and *Wnt4* [84]. Thus, DMRT1, on one hand, permits Sertoli cells to produce RA to support spermatogenesis and, on the other hand, it represses female promoting genes that would otherwise be active due to RA signaling. Generation of XY *Cyp26b1*-null embryos, in which endogenous RA is not degraded, confirmed that RA antagonizes testis development in mice [85]. Moreover, ectopic expression of *Dmrt1* induces XX sex reversal when it is produced before the sex determination stage [86], and reprogramming of juvenile and adult granulosa cells into Sertoli-like cells if it occurs after sex determination [87], indicating that the gene is not only necessary, but also sufficient to maintain the male sex fate in the supporting cells of the gonad.

### 4.2. SOX9/SOX8

SOX9 is a transcription factor belonging to the SOX [*Sry*-related high-mobility group (HMG) box] family that causes XY sex reversal in humans carrying a mutant allele of the gene [88,89]. It is expressed in Sertoli cells throughout life, and inactivation of the gene in mice before sex determination leads to XY sex reversal [90,91]. In contrast, mice with a Sertoli cell-specific ablation of *Sox9* shortly after the sex determination stage showed normal embryonic testis development and were initially fertile, but became sterile due to dysfunctional spermatogenesis at about five months [92]. SOX8 is a transcription factor that, together with SOX9 and SOX10, form the SOXE subgroup of proteins, which is also expressed in Sertoli cells throughout life [93]. Ablation of *Sox9* in Sertoli cells after the sex determination stage on a *Sox8^-/-^* background led to primary infertility with impairment of testis cord development at embryonic stages, proving redundant functionality between both SOXE genes in Sertoli cell differentiation [92]. In these double *Sox9/Sox8* mutants, upregulation of female promoting genes including *Rspo1*, *Wnt4*, and *Foxl2* and downregulation of *Dmrt1* are seen shortly after *Sox9* ablation [92,94], and gonadal transcriptome is completely feminized at P6 [95]. This functional redundancy between both SOXE genes also operates at the adult stage. Adult Sertoli cell-specific deletion of *Sox9* on a *Sox8^-/-^* background led to ectopic FOXL2 expression, Sertoli-to-granulosa cell transdifferentiation, and testis-to-ovary genetic reprogramming. This process of testis involution continues in time and finally leads to the complete degeneration of the seminiferous tubules, which become acellular, empty spaces among the extant Leydig cells [96]. In double *Sox9/8* mutant testes, DMRT1 protein only persists in non-mutant cells, showing that SOX9/8 are necessary to maintain *Dmrt1* expression in the adult testis and that Sertoli-to-granulosa cell transdifferentiation is mediated by *Dmrt1* downregulation in the absence of *Sox9/8* [96].

## 5. Antagonism between Male and Female Factors in Sexual Cell Fate Maintenance

Sex determination involves not only the activation of genes necessary for the differentiation of a sexual fate, but also the active repression of the genetic program of the opposite sex [16,17,18,19,20,22,23]. Some of these molecular interactions operate throughout life, and are summarized in Figure 2. The SOX9/SOX8-DMRT1 axis plays an central role in the maintenance of the Sertoli cell fate. Several studies have indicated that SOX9/SOX8 may control *Dmrt1* expression. Sertoli cell-specific ablation of *Sox9/8* at E13.5, shortly after the sex determination stage, leads to a rapid downregulation of *Dmrt1* which is observable just four days later, at E17.5 [94]. In contrast, in mice with a Sertoli cell-specific deletion of *Dmrt1*, *Sox9* is downregulated much later, at P14, coinciding with *Foxl2* upregulation [81]. This suggests that, whereas *Dmrt1* expression seems to be dependent of SOX9, *Sox9* expression is independent of DMRT1 and that the loss of *Sox9* after *Dmrt1* ablation is a secondary consequence of *Foxl2* upregulation. In contrast, other results suggest that DMRT1 might regulate *Sox9* expression as DMRT1 binds near the *Sox9* locus in P28 mouse testes [81]. In any case, we currently know that the three transcription factors cooperate in the maintenance of the Sertoli cell fate, as Sertoli-to-granulosa transdifferentiation in the postnatal testis is faster and more efficient when *Dmrt1*, *Sox9*, and *Sox8* are deleted in Sertoli cells, compared to single deletion of either *Dmrt1* or *Sox8/9* alone [84,97]. A recent study identified genes with Sertoli- and granulosa-biased postnatal expression and showed that many of them were associated with sex-biased differentially-accessible chromatin regions (DARs). In postnatal Sertoli cells, many of the Sertoli-biased DARs were bound by both DMRT1 and SOX9, confirming again postnatal cooperation between the two transcription factors in maintaining the Sertoli cell fate. Furthermore, ChIP-seq analysis of granulosa cells ectopically expressing *Dmrt1* or *Sox9* indicated that DMRT1 and SOX9 jointly bind many sites, although SOX9 was unable to bind most of these sites in the absence of DMRT1, suggesting that during the transdifferentiation process, DMRT1 acts as a pioneer factor promoting chromatin accessibility at regions where SOX9 binds subsequently [97]. Nevertheless, the fact that ectopic SOX9 expression can reprogram sex-biased gene expression in vitro without activating *Dmrt1* indicates that DMRT1-independent actions of SOX9 also exist during the Sertoli-to-granulosa transdifferentiation process [97]. Activation of the SOX9/SOX8-DMRT1 axis is essential to maintain *Foxl2* repressed, a process that is mediated by the antagonism of DMRT1 on the feminizing action of retinoic acid (RA) [81,94]. Two independent genetic cascades are necessary for embryonic and early postnatal maintenance of the female fate, one controlled by WNT signaling and the other by FOXL2 [98,99]. In the adult ovary, FOXL2 alone is essential to maintain the granulosa cell fate, whereas the involvement of WNT genes in female sex maintenance remains to be elucidated [37]. FOXL2 cooperates with ERα and ERβ in maintaining the granulosa cell fate by directly repressing the *Sox9* promoter [37,64]. In addition, the action of TGF-β, FST and LATS1/2, RUNX1, FOXO1/3 is necessary for maintaining the granulosa cell fate. Although it is known (1) that FOXL2 can cooperate with members of the TGF-β pathway in maintaining *Fst* expression [100,101]; (2) that FOXL2 is phosphorylated by LATS1, a process that seems to be important for granulosa cell differentiation and follicle maturation [102]; and (3) that FOXL2 and RUNX1 exhibit overlaps in chromatin binding in fetal ovaries [65], most of the molecular mechanisms governing the functional relationship among these female promoting genes remain unknown.

Finally, we would like to mention that alterations of the balance between male- and female-promoting factors are associated with gonadal diseases, particularly sex cord tumors. For example, a mutation of FOXL2 (C134W) is found in more than 97% of adult-type granulosa cell tumors [103]. In addition, other molecules and pathways such as inhibins, TGF-β, WNT, and SOX9 are associated with the pathogenesis or prognosis of gonadal tumors [53,104,105]. Thus, the exact knowledge of how the sex is maintained is important to understand gonadal function under normal and pathological conditions.

## Figures and Tables

**Figure 1 genes-12-00999-f001:**
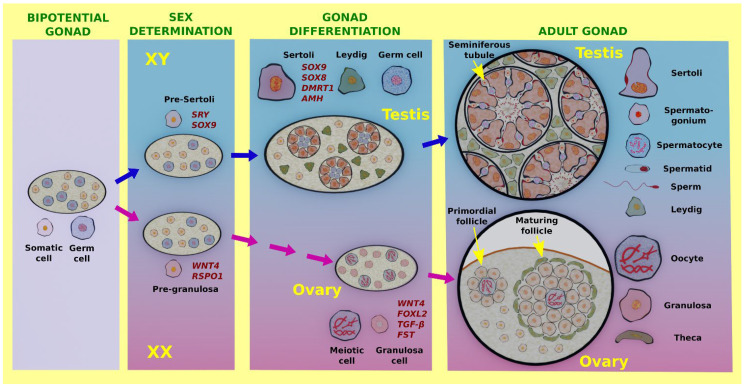
Gonadal cell-fate during gonad differentiation in mice. Prior to the sex determination stage, the bipotential gonad is composed of undifferentiated somatic cells (light pink) and germ cells (light blue). Sex differentiation starts with the specification (commitment) of the supporting cell progenitors to differentiate as either Sertoli cells in the testis or granulosa cells in the ovary. At the sex determination stage in XY individuals, the testis determining gene, *SRY*, starts to be expressed in the progenitors of the Sertoli cells (pre-Sertoli cells), leading to *SOX9* upregulation and Sertoli cell specification. Subsequently, pre-Sertoli cells undergo a mesenchymal to epithelial transition and differentiate into Sertoli cells that form the testis cords enclosing the XY germ cells. Sertoli cells follow a male-specific genetic program expressing genes such as *SOX9*, *SOX8*, *DMRT1*, and *AMH* promote the differentiation of the steroidogenic Leydig cells and prevent XY germ cells from meiosis entry. In mice, all these events are completed within 24–48 h after sex determination. In the adult testis, cords become seminiferous tubules with lumen and germ cells at different pre-meiotic, meiotic, and post-meiotic stages including spermatogonia, spermatocytes, spermatids, and sperm. In XX gonads, the male pathway is not activated at the sex determination stage due to the lack of *SRY* and the WNT signaling genes, *WNT4* and *RSPO1*, are upregulated in pre-granulosa cells. In the mouse, ovary differentiation is delayed with respect to testis differentiation and starts with meiosis initiation by XX germ cells, concomitant with the expression of further ovarian genes and pathways including *FOXL2*, TGFβ, and *FST*. Folliculogenesis is completed after birth, when germ cells are surrounded by a layer of granulosa cells. Follicle maturation in the mouse ovary begins a few days after birth including the proliferation of granulosa cells and the formation of an outer layer of steroidogenic theca cells.

**Figure 2 genes-12-00999-f002:**
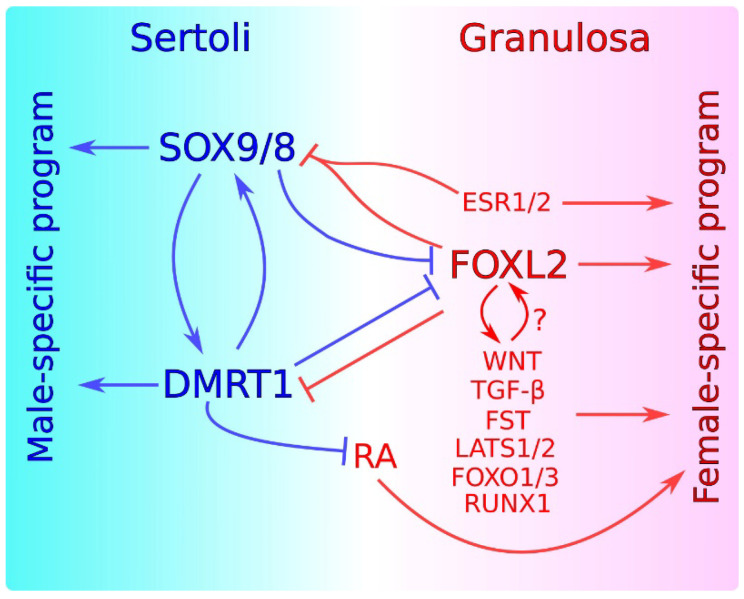
Current model for the maintenance of sex-specific supporting cell fates in adult gonads. In Sertoli cells, SOX9/8 establish a feed-forward regulatory loop with DMRT1, necessary for the maintenance of the male-specific program and for preventing the expression of ovary-promoting genes including FOXL2. DMRT1 inhibits RA signaling, which induces the expression of ovarian genes. In granulosa cells, FOXL2 interacts with ESR1/2 and probably with other genes and molecular pathways including WNT, TGFβ, FST, LATS1/2, FOXO1/3, and RUNX1 to maintain the female-specific program. FOXL2 together with ESR1/2 negatively regulates SOX9/8 and/or DMRT1. Male- and female-promoting genes are in blue and red, respectively. Blue and red lines represent an action exerted by male- and female-promoting genes, respectively. Positive regulation is indicated by arrows. Negative regulation is indicated by perpendicular lines.

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
