# Peer review of "Sex Maintenance in Mammals"

_genes, 2021, doi:10.3390/genes12070999_

Round 1

Reviewer 1 Report

This is a timely review which is focussed on events in mammals that occur after the gonads have formed. A body of evidence is nicely assembled showing that the fully differentiated gonads are not locked into their fate but are malleable. Key to this understanding are the insights that show us the genes that are required to maintain a fully differentiated testis or ovary. In particular, the loss of either FOXL2 or DMRT1 can result in trans-differentiation into the opposite sex gonad. This confirms that a suit of genes is required to maintain the differentiated gonads in their state as a testis or ovary. The review is comprehensive in marshalling the evidence for this thesis and will be invaluable to all with an interest in this field and beyond.

Author Response

We thank the constructive comments of the reviewers, which helped us to improve the manuscript.

Reviewer 1

The English has been revised thoroughly and several changes have been done in order to improve it.

Reviewer 2 Report

In this review, the authors summarize current understanding of the genetic events involved in promoting and preserving the differentiation of the female versus the male gonad. These genetic data are mainly based on results obtained in the mouse species.

Major points:

1. The authors mix information on transdifferentiation process (differentiation of Sertoli-like cells in ovary or granulosa-like cells in testis) and tumorigenesis (Sertoli cell tumour or Granulosa cell tumour). These processes are different and make the manuscript unclear and confused.

All references to tumours have to be removed from the main text as these have nothing to deal with the initial question which aim to understand the “battle of the sex” and the plasticity of the gonadal epigenome that allows switching from one programme to the other.

- Line 219-223 on TGFb and tumorigenesis.

- Paragraph 3.5 (line 292-302) on FOXO1 & 3. 

- Paragraph 3.6 (line 303-317) on Hippo signalling.

If tumorigenesis process needs to be discussed, these elements could be used in the last part of the review where the authors want to open the discussion on sex-tumours (conclusion part).

Another option could be to add as specific paragraph on this topic, in order to better understand how granulosa-cell or sertoli cell tumor analyses may increase our knowledge or understanding of sex determination and/or maintenance.

2. The authors have forgotten important data on FOXL2 coming from the goat species. Indeed FOXL2 knock-out in XX gonad induces female to male sex reversal with the differentiation of XX testis (XX males) [PMID: 24485832, PMID: 25395674]. While in the mouse FOXL2 is the gatekeeper of the ovarian (granulosa cell) identity afterbirth, in the goat, FOXL2 is involved in the ovarian determination very early in the fetal life.

This aspect has to be discussed in the manuscript, or the title couldn’t be sex maintenance in “mammals”.

FOXL2 counterpart in the testis is DMRT1. While in the mouse DMRT1 is the gatekeeper of the testis (Sertoli cell) identity after birth, in humans DMRT1 is involved in testis determination and may be necessary to activate/regulate SOX9.

 A recent publication from D. Zarkower lab [PMID: 24485832] showed in murine post-natal Sertoli cells that DMRT1 is necessary for SOX9 action as DMRT1 may act as a pioneer factor opening the chromatin and promoting  SOX9 binding to its specific sites. I think these new data are worth to be discussed in the present manuscript.  

Minor points:

  1. Line 43-44 : “XX-XY chimaeric mouse testes showed that Sertoli cells were exclusively XY”. This affirmation is not correct. They actually showed that about 10% of Sertoli cells were XX [PMID: 1769333]. This aspect shows that XX supporting cells, that cannot express SRY, are secondary recruited via paracrine signals.
  2. Figure 1. “Gonadal cell-fate during gonad differentiation”... Please specify “in mice”.
  3. Line 109. Typo : “are upregulated” rather than “unregulated”.
  4. Line 109/110 : “ovarian differentiation is delayed with respect to testis differentiation and starts with meiosis”. If the authors are talking about cellular organisation, this is only true in the mouse species (not the case in humans, cattle, sheep, rabbit etc…). Mice present an “immediate meiosis” while the other mammals present a “delayed meiosis” and harbour ovigerous cords formation prior to meiosis initiation. In addition, even in the mouse, the ovarian differentiation is not delayed when looking at the molecular level [PMID: 25158167 ]. This sentence must be modified.
  5. Line 112-113: “Follicle maturation begins at puberty and includes the formation of an outer layer of thecal cells”. This is not correct. At puberty it is the terminal part of follicle maturation that occurs, the one that is dependant of gonadotrophins. In the mouse ovary, few days after birth, first waves of follicle growth are observed (until early antral stage). Thecal cells first appear surrounding secondary follicles (thus not only at puberty). In addition the “follicle” presented on figure 1 has one layer of cells and should represent a primary follicle that can’t have surrounding thecal cells. Please modify the text and the figure.
  6. Line 166-170. Here again there is a confusion between tumours formation and transdifferentiation process. It is underlined that this process is not related to germ cell loss, but : (i) here we are not talking about transdifferentiation process, but rather of tumours formation that harbour markers of granulosa-cell tumours and (ii) germ cell loss in a testis has never been associated with transdifferentiation process and germ cells are not necessary to testis cords formation. This sentence has to be removed or rewritten.
  7. Line 185-189. The sentence has to be rewritten in a more positive/affirmative way: H3K27me3 indicates promoter repression and H3K4me3 promoter activation. ChIP-seq experiments showed that male-promoting genes harboured H3K4me3 and are depleted in H3K27me3 in adult Sertoli cells, while female-promoting genes are enriched for both marks….

Author Response

We thank the constructive comments of the reviewers, which helped us to improve the manuscript.

Reviewer 2

Major points:

1. The authors mix information on transdifferentiation process (differentiation of Sertoli-like cells in ovary or granulosa-like cells in testis) and tumorigenesis (Sertoli cell tumour or Granulosa cell tumour). These processes are different and make the manuscript unclear and confused.

All references to tumours have to be removed from the main text as these have nothing to deal with the initial question which aim to understand the “battle of the sex” and the plasticity of the gonadal epigenome that allows switching from one programme to the other.

- Line 219-223 on TGFb and tumorigenesis.

- Paragraph 3.5 (line 292-302) on FOXO1 & 3.

- Paragraph 3.6 (line 303-317) on Hippo signalling.

If tumorigenesis process needs to be discussed, these elements could be used in the last part of the review where the authors want to open the discussion on sex-tumours (conclusion part).

Another option could be to add as specific paragraph on this topic, in order to better understand how granulosa-cell or sertoli cell tumor analyses may increase our knowledge or understanding of sex determination and/or maintenance.

Response: We just mentioned that deregulation of some of these factors cause gonadal tumour as an introductory description of the gonadal phenotype, and subsequently we focused in the sexual transdifferentiation process. But we understand the reviewer’s concern that such descriptions may distract the reader's attention from the main topic of this manuscript. Accordingly, we have made the following changes:

Lines 225-226 (revised version of the manuscript): we removed “, and double-homozygous Inhibin and FSH mutant ovaries contained Sertoli cell tumors”.

Lines 228-230: We substituted “...in Sertoli cells also led to granulosa cell tumors whose cells expressed Inhnibin A and Foxl2 and upregulated the Wnt signaling pathway” with “ in Sertoli cells led to Inhibin A, FOXL2, and WNT signaling upregulation”.

Lines 300-301: We substituted “the penetrance and onset of ovary tumors formation” with “the penetrance of this phenotype”.

Line 309: We substituted “no corpora lutea, and late onset of stromal tumors” with “and no corpora lutea.

2. The authors have forgotten important data on FOXL2 coming from the goat species. Indeed FOXL2 knock-out in XX gonad induces female to male sex reversal with the differentiation of XX testis (XX males) [PMID: 24485832, PMID: 25395674]. While in the mouse FOXL2 is the gatekeeper of the ovarian (granulosa cell) identity after birth, in the goat, FOXL2 is involved in the ovarian determination very early in the fetal life.

This aspect has to be discussed in the manuscript, or the title couldn’t be sex maintenance in “mammals”.

Response: We agree with the reviewer and, accordingly, we have inserted the following sentence in lines 260-265: “The first evidence that this factor could play a role in mammalian sex determination came from the identification of a deletion of 11.7 kb in the sex-reversed polled goat, which included FOXL2 (Pailhoux, 2001). A later study showed that FOXL2 is a female sex-determining gene in the goat (Boulanger, 2014). In contrast, FOXL2 is dispensable for mouse sex determination, although it has essential roles during ovarian development”.

FOXL2 counterpart in the testis is DMRT1. While in the mouse DMRT1 is the gatekeeper of the testis (Sertoli cell) identity after birth, in humans DMRT1 is involved in testis determination and may be necessary to activate/regulate SOX9.

 A recent publication from D. Zarkower lab [PMID: 24485832] showed in murine post-natal Sertoli cells that DMRT1 is necessary for SOX9 action as DMRT1 may act as a pioneer factor opening the chromatin and promoting  SOX9 binding to its specific sites. I think these new data are worth to be discussed in the present manuscript.  

This paper came out after our initial submission We were also aware of it, and we agree with the review that these results must be mentioned in the present review. Accordingly, we have included a paragraph in lines

Response: This paper came out after our initial submission. We were also aware of it and we agree with the reviewer that these results must be mentioned in our review. Accordingly, we have included the following paragraph in lines 400-417: In any case, we currently know that the three transcription factors cooperate in the maintenance of the Sertoli cell fate, as Sertoli-to-granulosa transdifferentiation in the postnatal testis is faster and more efficient when Dmrt1, Sox9, and Sox8 are deleted in Sertoli cells, compared to single deletion of either Dmrt1 or Sox8/9 alone (Minkina 2014; Lidenman 2021). A recent study identified genes with Sertoli- and granulosa-biased postnatal expression and showed that many of them are associated with sex-biased differentially-accessible chromatin regions (DARs). In postnatal Sertoli cells, many of the Sertoli-biased DARs were bound by both DMRT1 and SOX9, confirming again postnatal cooperation between the two transcription factors in maintaining the Sertoli cell fate. Furthermore, ChIP-seq analysis of granulosa cells ectopically expressing Dmrt1 or Sox9, indicated that DMRT1 and SOX9 jointly bind many sites, although SOX9 was unable to bind most of these sites in the absence of DMRT1, suggesting that, during the transdifferentiation process, DMRT1 acts as a pioneer factor promoting chromatin accessibility at regions where SOX9 binds subsequently (Lidenman 2021). Nevertheless, the fact that ectopic SOX9 expression can reprogram sex-biased gene expression in vitro without activating Dmrt1, indicates that DMRT1-independent actions of SOX9 also exist during the Sertoli-to-granulosa transdifferentiation process

Minor points:

Line 43-44 : “XX-XY chimaeric mouse testes showed that Sertoli cells were exclusively XY”. This affirmation is not correct. They actually showed that about 10% of Sertoli cells were XX [PMID: 1769333]. This aspect shows that XX supporting cells, that cannot express SRY, are secondary recruited via paracrine signals.

Response: The reviewer is right. In lines 44-46, we have substitutedSertoli cells were exclusively XY, whereas XX cells contributed to other cell types,withSertoli cells were predominantly XY, whereas XX cells contributed mainly to other cell types

Figure 1. “Gonadal cell-fate during gonad differentiation”... Please specify “in mice”.

Response: Done

Line 109. Typo : “are upregulated” rather than “unregulated”.

Response: Done

Line 109/110 : “ovarian differentiation is delayed with respect to testis differentiation and starts with meiosis”. If the authors are talking about cellular organisation, this is only true in the mouse species (not the case in humans, cattle, sheep, rabbit etc…). Mice present an “immediate meiosis” while the other mammals present a “delayed meiosis” and harbour ovigerous cords formation prior to meiosis initiation. In addition, even in the mouse, the ovarian differentiation is not delayed when looking at the molecular level [PMID: 25158167 ]. This sentence must be modified.

Response: The reviewer is right and we are also aware of the fact that mice are exceptional in this respect (see Jiménez 2009; PMID: 20130386). That is why our original sentence in line 115 states “In the mouse, ovary differentiation is delayed with respect to testis differentiation and starts with meiosis initiation by XX germ cells”. So, no correction is needed here.

Line 112-113: “Follicle maturation begins at puberty and includes the formation of an outer layer of thecal cells”. This is not correct. At puberty it is the terminal part of follicle maturation that occurs, the one that is dependant of gonadotrophins. In the mouse ovary, few days after birth, first waves of follicle growth are observed (until early antral stage). Thecal cells first appear surrounding secondary follicles (thus not only at puberty). In addition the “follicle” presented on figure 1 has one layer of cells and should represent a primary follicle that can’t have surrounding thecal cells. Please modify the text and the figure.

Response: The reviewer is right. We have modified both the text and the figure, to be more precise in this description. In lines 118-121, we substituted “Follicle maturation begins at puberty and includes the formation an outer layer of theca cells” with “Follicle maturation in the mouse ovary begins a few days after birth, including the proliferation of granulosa cells and the formation of an outer layer of steroidogenic theca cells

Line 166-170. Here again there is a confusion between tumours formation and transdifferentiation process. It is underlined that this process is not related to germ cell loss, but : (i) here we are not talking about transdifferentiation process, but rather of tumours formation that harbour markers of granulosa-cell tumours and (ii) germ cell loss in a testis has never been associated with transdifferentiation process and germ cells are not necessary to testis cords formation. This sentence has to be removed or rewritten.

Response: In this paragraph we just wanted to put an example of the presence of granulosa cell in the adult testis. Some cases of young granulosa cell tumors have an intratubular component suggesting that neoplastic proliferation of intratubular sex cord cells progresses to an invasive tumor, simultaneously acquiring granulosa cell differentiation and losing Sertoli cell features, including increasing FOXL2 and decreasing SOX9 reactivity (Kao et al., 2015; PMID: 26076062). However, in most of the cases, as the reviewer indicates, granulosa cell tumors are testis tumors that express some granulosa cell markers. Regarding germ cells the reviewer is right. Accordingly, in lines 174-183, we have substituted the sentences “On the other hand, the granulosa-cell tumor of the testis is another type of rare tumor in which granulosa cells, and occasionally theca cells, are present in male patients [39,40]. In these cases, transdifferentiation is not associated to germ cell loss, and it is believed that a disruption in the the balance between sex-specific determining signals is the underlying molecular cause of this cancer.” with these ones: On the other hand, cases of granulosa-cell tumors have also been reported in which neoplastic proliferation of intratubular sex cord cells progresses to an invasive tumor, simultaneously experiencing granulosa cell differentiation and losing Sertoli cell features. However, in both cases supporting cell transdifferentiation may be just a secondary consequence of the dramatic alterations taking place in the genetic program of tumor cells.

Line 185-189. The sentence has to be rewritten in a more positive/affirmative way: H3K27me3 indicates promoter repression and H3K4me3 promoter activation. ChIP-seq experiments showed that male-promoting genes harboured H3K4me3 and are depleted in H3K27me3 in adult Sertoli cells, while female-promoting genes are enriched for both marks….

Response: We acknowledge that our original sentence was not easily readable. According to the reviewer suggestions, we substituted it with this one: “ChIP-seq for H3K27me3 and H3K4me3 provided results consistent with this notion. H3K27me3 indicates promoter repression, whereas H3K4me3 evidences promoter activation. ChIP-seq experiments performed using purified adult Sertoli cells showed that male-promoting genes harboured H3K4me3 and were depleted in H3K27me3, whereas female-promoting genes were enriched for both marks, indicating that female-determining genes persist in a poised state even long after Sertoli cell differentiation